# A Contactless Laser Doppler Strain Sensor for Fatigue Testing with Resonance-Testing Machine

**DOI:** 10.3390/s21010319

**Published:** 2021-01-05

**Authors:** Fangjian Wang, Steffen Krause, Joachim Hug, Christian Rembe

**Affiliations:** 1Institute of Electrical Information Technology, Clausthal University of Technology, Leibnizstr. 28, D-38678 Clausthal-Zellerfeld, Germany; fangjian.wang@tu-clausthal.de; 2SincoTec Test & Engineering GmbH, Innovationspark Tannenhöhe, Freiberger Str. 13, D-38678 Clausthal-Zellerfeld, Germany; krause@sincotec.de; 3SincoTec Holding GmbH, Innovationspark Tannenhöhe, Freiberger Str. 13, D-38678 Clausthal-Zellerfeld, Germany; joachim.hug@sincotec.de

**Keywords:** dynamic strain measuring, remote sensing, in-plane laser Doppler vibrometry

## Abstract

In this article, a non-contact laser Doppler strain sensor designed for fatigue testing with the resonance-testing machine is presented. The compact sensor measures in-plane displacements simultaneously from two adjacent points using the principle of in-plane, laser-Doppler vibrometry. The strain is computed from the relative displacements divided by the distance between these two points. The optical design, the mathematical model for estimating noise-limited resolution, the simulation results of this model, and the first measurement results are presented. The comparison of the measurement results of our sensor with the results of a conventional strain gauge shows that our design meets the measurement requirements. The maximum strain deviation compared to conventional strain gauges of the laser-Doppler extensometer is below $4\times 10^{-5}$ in all performed experiments.

## 1. Introduction

In fatigue testing, strain gauges are widely used to measure strain. Even though these traditional sensors are inexpensive and technically well developed, they have some disadvantages. An additional measuring error, which is not included in the uncertainty specifications of the strain-gauge manufacturer, results from the elasticity of the adhesive material that must be applied between the strain gauge and the measuring surface. Sensitivity alters with changing temperatures at the measuring point [1]. The influences of displacements transverse to the direction of strain measurement cannot be avoided. A zero signal drift occurs as a characteristic feature, resulting from fatigue damage caused by dynamic stress in the strain gauge after a certain number of load cycles [1]. A measurement configuration with many measurement points also causes considerable cost and effort during setup. The quality of the installation greatly influences the accuracy and reliability of the measurement [1].

The use of fiber optic strain sensors with Bragg gratings [2] enables better accuracy at the sub-microstrain level. Fabricated in polarization-maintaining fiber, the strain resolution of the fiber Bragg gratings sensor can be pushed below $10^{-9}\;1/\sqrt{\mathrm{Hz}}$ [3]. However, the coupling problem of tactile strain measurement still exists. This problem can be solved with contactless optical measurement methods. Camera-based measuring devices such as digital image correlation [4,5,6,7], electronic speckle interferometry [8,9], and laser speckle correlation [10,11] are widely used for strain measurement, even with special measuring conditions like measuring strain under high temperature [12,13]. A maximum strain error of about $3\times 10^{-5}$ can be reached, but only with a low sampling rate of about 100 Hz due to the camera image frequency and the time-consuming calculation of image processing [6,7]. Shearography, as an interferometric technique, can also be applied for in-plane displacement or strain measurement [14,15,16]. The common implementation with the Michelson interferometer as the shearing device limits the light efficiency up to 50%. Similar to other speckle interferometric techniques, shearography has limited tolerance to rigid body motion including object translation. At high image repetition rates, image processing usually does not produce a real-time signal. Therefore, such methods are not suitable for dynamic strain measurement, especially in a high-frequency range. Moreover, the maximal measurable displacements of these methods as well as strain gauges are limited. In the case of 3D-scanning laser Doppler vibrometry [17,18,19], scanning also results in a long measuring time (about 30 scan points per second), so that no real-time signal of the strain can be obtained. Such methods fit better for full-field strain measurement. Spatial-filtering sensors [20,21] employ the detection of spatial frequency components with fast photodiodes by means of apertures. A corresponding sensor provides a real-time signal as a result but the spatial filterwhich has not a pure sine transfer behavior (usually due to its binary transmittance [22]), causes phase distortion of the detector signals and consequently distorts the displacement signal that is demodulated from the detector signals [20]. Besides, only a part of the scattered light is used since the light through the receiving lens system with a large numerical aperture is spatial filtered. Therefore, the signal-to-noise ratio (SNR) is reduced.

Compared with other optical methods, laser Doppler velocimetry [23] has the advantages of very high time resolution and low cost of data processing. It measures in-plane velocity by calculating the Doppler frequency shift and in-plane displacement via the calculating phase (integration of the Doppler frequency shift). M. Hercher demonstrated a laser Doppler technique for extensometers in 1987 [24] and astonishingly no further work on this method has been published, although strain measurements with two separate in-plane laser Doppler vibrometers are state-of-the-art. However, in-plane laser Doppler vibrometers were explored by other groups. As a result of these publications, reflective power and surface roughness are the parameters that mostly influence the accuracy [25,26]. Later, the measuring accuracy of in-plane displacement was improved to the sub-micrometer area [27]. By rotating the direction of incident beams and receiving optics, even the in-plane motion of objects with a glossy and specular surface can be measured using a laser Doppler technique [28].

We wondered why the approach of Hercher was not further investigated since 1987 and explored if this technique can be realized as a single sensor with one beam path, whereby the sensor analyzes two measuring points simultaneously. Assuming that the technique was not advanced further due to physical limitations, we developed a sophisticated model to explore the physics of this sensor principle. Our simulations convinced us that strain resolutions in the range of classical strain gauges can be achieved. The model allowed us to find optimal design parameters for a compact laser Doppler strain sensor for the first time. In a first conference paper, we demonstrated the first preliminary results [29] showing the feasibility of the approach. In this paper, we demonstrate a robust and compact laser Doppler extensometer suitable for industrial applications approaching for the first time the resolution of classical strain gauges of approximately $10^{-5}$. The sensor was integrated into a resonance-testing machine (POWER SWINGly 20 kN from SincoTec) to prove its suitability as a strain gauge replacement. We prove in this paper that our model has resulted in design parameters enabling strain resolutions below $10^{-5}$ for a resolution bandwidth of 2.37 Hz and maximal strain deviations of $4\times 10^{-5}$ compared to a conventional strain gauge.

## 2. Optical Setup

The optical layout of our new remote strain sensor is shown in Figure 1. A near-infrared (NIR) laser (1550 nm) is used as the light source, which allows 10 times more light power than in the visible range for an eye-safe laser class sensor. A polarizing beam splitter (PBS) divides the incident beam into two parts. The s-polarizing component is deflected to the surface of the specimen as one measurement beam (Beam A). A Bragg cell shifts the frequency of the p-polarized ray by 40 MHz and turned its polarization to perpendicular, which is also deflected to the specimen (Beam B). Before reaching the specimen surface, a non-polarizing beam splitter (BS) and a prism divided each incident ray into two parallel beams (Beam 1 into Beam A1 and A2, and Beam B as well). Beam A1 and Beam B1 are superimposed on the surface of the object and create an interference fringe pattern, which is the first measuring point. Similarly, Beam A2 and B2 are superimposed to define the second measuring point.

By each measuring point, the frequency of the intensity modulation of the fringe pattern changes proportionally to the velocity of a scattering particle from the rough surface that moves through the pattern. The phase of the intensity modulation is then proportional to the displacement. Usually, several particles move through the fringe pattern and created a time-dependent speckle-pattern. Then a lens system collects the diffracted light scattered at both measuring points. The light from each measuring point is imaged onto a single photodetector. The in-plane displacement could be derived by analyzing the intensity modulation phase that corresponded to the integration of the Doppler-frequency shift. The modulation frequency is shifted by an in-plane deformation in the direction perpendicular to the interference fringes of the fringe pattern on the surface of the specimen. The measuring point running against the interference fringes increases the Doppler-shift frequency (difference in frequency between intensity modulation and 40 MHz carrier frequency) and running with the fringes reduces the Doppler-shift frequency. Here, the Doppler-shift frequencies are rather derivatives of phases $\phi_{i}\left(t\right)$ of intensity modulation from the i-th measuring position and are only fixed frequencies at constant velocities. Therefore, it is possible to detect the direction of movements. The measured strain $\varepsilon^{\prime }\left(t\right)$ is then given by
(1)$$\varepsilon^{\prime }\left(t\right)=\frac{d}{2\pi l}\int_{t_{0}}^{t}\left({\dot{\phi }}_{1}\left(\tilde{t}\right)-{\dot{\phi }}_{2}\left(\tilde{t}\right)\right)d\tilde{t}=\frac{d}{2\pi l}\left(\phi_{1}\left(t\right)-\phi_{1}\left(t_{0}\right)-\phi_{2}\left(t\right)+\phi_{2}\left(t_{0}\right)\right),$$
where d is the distance between two nearby fringes, l is the distance between the center of two measuring points, and $t_{0}$ is the starting time. The fringe pattern with the image area of the photodiode is shown in Figure 2.

In addition, the strain measured by our optical sensor $\varepsilon^{\prime }\left(t\right)$ from Equation (1) is not exactly the physical strain $\varepsilon \left(t\right)$, which is defined by
(2)$$\varepsilon \left(t\right)=\frac{\Delta l\left(t\right)}{l},$$
where l is the original length of the test specimen and $\Delta l\left(t\right)$ is the change in length. Now we define the velocities of both endpoints of the test specimen as $v_{1,p}\left(t\right)$ and $v_{2,p}\left(t\right)$ (in Figure 3a).

The original distance between both these endpoints is equal to the length of the measuring object l and the actual distance is then l+∆l. The differential of the physical strain $\dot{\varepsilon }\left(t\right)$ is proportional to the difference of the velocities at both endpoints $v_{1,p}\left(t\right)$ and $v_{2,p}\left(t\right)$.
(3)$$\dot{\varepsilon }\left(t\right)=\frac{d\frac{\Delta l\left(t\right)}{l}}{dt}=\frac{v_{1,p}\left(t\right)-v_{2,p}\left(t\right)}{l}$$We consider the distance between two measuring points of the laser Doppler strain sensor is also equal to l. The velocities measured by the laser Doppler strain sensor could be defined as $v_{1,L}$ and $v_{2,L}$ (in Figure 3b), which could be computed from the Doppler-shift frequencies $f_{i}\left(t\right)={\dot{\phi }}_{i}\left(t\right)/2\pi$ and the fringe distance d.
(4)$$v_{i,L}=\frac{d}{2\pi }{\dot{\phi }}_{i}\left(t\right)$$The measured strain $\varepsilon^{\prime }\left(t\right)$ from Equation (1) is then given by
(5)$$\varepsilon^{\prime }\left(t\right)=\frac{1}{l}\int \left(v_{1,L}\left(t\right)-v_{2,L}\left(t\right)\right)dt.$$Considering the homogeneous material, the relationship between the velocities of both endpoints of the measuring object $v_{1,p}$, $v_{2,p}$ and the velocities measured by the laser Doppler strain sensor $v_{1,L}$, $v_{2,L}$ is given by
(6)$$\frac{v_{1,L}\left(t\right)-v_{2,L}\left(t\right)}{v_{1,p}\left(t\right)-v_{2,p}\left(t\right)}=\frac{l}{l+\Delta l\left(t\right)}.$$With (6) in (5):(7)$$\varepsilon^{\prime }\left(t\right)=\frac{1}{l}\int \left(\frac{l}{l+\Delta l\left(t\right)}\;\left(v_{1,p}\left(t\right)-v_{2,p}\left(t\right)\right)\right)dt$$With $v_{1,p}\left(t\right)-v_{2,p}\left(t\right)=l\cdot \dot{\varepsilon }\left(t\right)$ and $\Delta l\left(t\right)=l\cdot \varepsilon \left(t\right)$ in (7) the relation between the physical and the measured strain is derived.
(8)$$\begin{array}{c} \varepsilon^{\prime }\left(t\right)=\frac{1}{l}\int \left(\frac{l}{l+l\cdot \varepsilon \left(t\right)}\;\left(l\cdot \dot{\varepsilon }\left(t\right)\right)\right)dt\\ =\int \frac{\dot{\varepsilon }\left(t\right)}{1+\varepsilon \left(t\right)}\;dt\\ =\ln\left(1+\varepsilon \left(t\right)\right) \end{array}$$After reforming (8), the exact physical strain could be calculated from
(9)$$\varepsilon \left(t\right)=e^{\varepsilon^{\prime }\left(t\right)}-1.$$This formula shows that a compensable difference of the measured strain $\varepsilon^{\prime }\left(t\right)$ compared with the physical strain $\varepsilon \left(t\right)$ increases with strain size. The Taylor expansion of the right-hand side of (9) shows that the physical and measured strain are equal for the first order of the Taylor expansion. Therefore, we could expect small measurement errors for small strains (<1%).

## 3. Modeling

In this part, a mathematical model is presented to estimate the theoretical resolution limit of our sensor. The relationship between the resolution limit and the crossing angle θ, the angle between the incident laser beams, is studied.

### 3.1. Signal Power

The total scattered radiant power $\Phi_{s}$ depends on the incident radiant power $\Phi_{in}$. In addition, the incident power $\Phi_{in}$ is in relation to the incident irradiance $E_{in}$ so that the total scattered radiant power $\Phi_{s}$ is given by
(10)$$\Phi_{s}=\eta \Phi_{in}=\eta E_{in}A_{m}cos\frac{\theta }{2},$$
where η is the scattering efficiency, $\Phi_{in}$ is the incident radiant power,$\;E_{in}$ is the incident irradiance, $A_{m}$ is the area of the fringe pattern, and θ is the crossing angle. The scattered radiant power could also be calculated by
(11)$$\Phi_{s}=2\pi L_{s}A_{m}.$$With (10) and (11) the scattered radiance $L_{s}$ is given by
(12)$$L_{s}=\frac{\eta }{2\pi }E_{in}cos\frac{\theta }{2}.$$If we consider the measuring surface as a Lambertian surface, the scattered radiant intensity $I_{s}$ obeys Lambert’s cosine law: $I_{s}\left(\varphi^{\prime }\right)=L_{s}A_{0}cos\varphi^{\prime },$ where $A_{0}$ is the image area of the photodiode on the surface of the specimen and $\varphi^{\prime }\in \left(0,\varphi /2\right)$ is the observer angle. Therefore, the receiving light power of the photodiode $P_{D,i}$ from a single incident beam, which could be calculated with integration of the scattered radiant intensity $I_{s}$ via the solid angle Ω of the receiving optics, is given by
(13)$$\begin{array}{l} P_{D,single}=\int I_{s}d\Omega \\ =\pi L_{s}A_{0}\left(1-cos\varphi \right)\\ =\frac{1}{2}\eta E_{in}A_{0}\cos\frac{\theta }{2}\left(1-cos\varphi \right). \end{array}$$Considering the interference of two incident beams, the total scattered light power could be calculated by
(14)$$P_{D}=2P_{D,single}+2\;{\left(P_{D,single}^{2}\right)}^{\frac{1}{2}}\;cos\left(2\pi f_{c}t+\Delta \phi \right),$$
where $f_{c}=40\;MHz$ is the carrier frequency produced by the Bragg cell, and Δϕ is the phase. Only the AC component with a root mean square (RMS) value $\sqrt{2}P_{D,single}$ represent the power of the heterodyne carrier signal.
(15)$$P_{D,signal}=\sqrt{2}P_{D,single}\;,$$A photodiode collects this light and produces a photocurrent. With a trans-impedance amplifier, the photocurrent is converted to a voltage signal. Consequently, the quadratic RMS signal amplitude in voltage $u_{s}{}^{2}$ could be computed from
(16)$$u_{s}{}^{2}=\frac{1}{2}{\left(R_{f}\delta_{s}\delta_{I}KP_{D,signal}\right)}^{2},$$
where $R_{f}$ is the feedback resistance of the trans-impedance amplifier, $\delta_{s}$ is a factor in relation to the radiant intensity of the Lambert diffuser, and K is the sensitivity of the photodiode. In addition, $\delta_{I}$ is a factor for the consideration of the interference effect. The scattered light from simultaneous light-dark and dark-light transitions of the scattering bodies could destructively interfere (in Figure 4).

This leads to a deterioration of the carrier signal quality. *A* small fringe spacing d and a small crossing angle θ, therefore, have a detrimental effect on the SNR. We assume as an approximation for the factor $\delta_{I}$
(17)$$\delta_{I}=\frac{1}{\sqrt{\sqrt{A_{0}}/d}}=\frac{\sqrt{d}}{\sqrt[{4}]{A_{0}}}$$
with $d\leq \sqrt{A_{0}}$, where $A_{0}$ is the image area of the photodiode and $\sqrt{A_{0}}$ is the width of the quadratic image area, and $d=\lambda /2sin\frac{\theta }{2}$ is the fringe distance. Here λ is the incident wavelength and θ is the crossing angle. We justify our assumption (17) because only one fringe in the image area would result in a maximum factor $\delta_{I}=1$, and infinite fringes would result in the factor $\delta_{I}=0$. The behavior of this phenomenological introduced parameter is physically conclusive. If there is one maximal fringe in the image area, the factor of destructive interference $\delta_{I}$ is set to 1. Consequently, the factor $\delta_{I}$ is
(18)$$\delta_{I}=\left\{\begin{array}{c} \frac{\sqrt{d}}{\sqrt[{4}]{A_{0}}},\;\;|d\leq \sqrt{A_{0}}\\ 1,\;\;\;\;\;\;\;|d>\sqrt{A_{0}} \end{array}\right.$$If we change the crossing angle from 10° to 120°, the value of the factor $\delta_{I}$ varies from 0.24 to 0.08. All three factors, scattering efficiency η, a factor in relation to the radiant intensity of the Lambert diffuser $\delta_{s}$, and the factor for destructive Interference $\delta_{I}$ are estimated values. With them, the total efficiency is estimated between 2~6%. Only 2~6% of the incident power on the image area is received as the heterodyne signal by the photodiode. In the practice, the efficiency varies with different specimen surface. A more explicit description of the influence of the speckle effect and destructive interference may be researched later. Using Formulas (13), (15), and (16), the quadratic RMS signal amplitude in voltage $u_{s}{}^{2}$ from (16) is computed from
(19)$$u_{s}{}^{2}=\frac{1}{4}{\left(\pi^{2}R_{f}\delta_{s}K\eta \delta_{I}E_{in}A_{0}cos\frac{\theta }{2}\left(1-cos\varphi \right)\right)}^{2}.$$
with $\delta_{I}$ given by Formula (18).

### 3.2. Noise Analysis and SNR

For each measuring point, a single photodiode and a trans-impedance amplifier are used to detect the light signal. The common noise sources are thermal noise, voltage, and current noise from the amplifier. Here, because of little receiving light power (less than 0.1 mW) shot noise, which is much less than 1% of thermal noise, it could be neglected. The noise density of thermal noise is given by
(20)$$e_{nR}=\sqrt{4k_{B}TR_{f}},$$
where $k_{B}$ is Boltzmann’s constant, T is the temperature. The total noise density with negative input of the amplifier is computed from
(21)$$e_{ne}=\sqrt{e_{nR}{}^{2}+e_{n}{}^{2}+{\left(i_{n}R_{f}\right)}^{2}}$$
where $e_{n}$ is the input voltage noise density of the amplifier and $i_{n}$ is the input current noise density of the amplifier. The total noise density with the output of the amplifier is then given by
(22)$$e_{n,total}=G_{n}e_{ne},$$
where $G_{n}$ is the noise gain of the trans-impedance amplifier. Since the signal bandwidth is much smaller than the carrier frequency, we could consider the noise gain as a constant near the operating point. The total square noise voltage is calculated as
(23)$$u_{n}{}^{2}=e_{n,total}{}^{2}B,$$
where B is the bandwidth. The total SNR is then
(24)$$SNR=\frac{u_{s}{}^{2}}{u_{n}{}^{2}}=\frac{\frac{1}{4}{\left(\pi^{2}R_{f}\delta_{s}K\eta \delta_{I}E_{in}A_{0}cos\frac{\theta }{2}\left(1-cos\varphi \right)\right)}^{2}}{G_{n}^{2}\left(4k_{B}TR_{f}+e_{n}{}^{2}+{\left(i_{n}R_{f}\right)}^{2}\right)B}.$$

### 3.3. Noise Limited Resolution

If the minimum detectable displacement $s_{min}$ of the i-th measuring point could be detected, the phase deviation $\Delta \phi_{i}$ caused by this displacement should be above the phase noise $\Delta \phi_{n}$. Here, the phase $\phi_{i}$ is given by the sinusoidal carrier signal $\sqrt{2}u_{s}sin\phi_{i}$. The phase deviation $\Delta \phi_{i}$ has a maximum rise $\pm \sqrt{2\;}u_{s}$, when a scattering body crosses the light-dark or dark-light transition (in Figure 5).

In this case, the phase deviation $\Delta \phi_{i}$ is given by
(25)$$\Delta \phi_{i}=2\pi \frac{s_{min}}{d}\;,$$
with the fringe distance d. Equivalently, the minimum detectable signal amplitude deviation caused by this phase deviation is equal to the voltage deviation $u_{n}$ caused by the total noise. Because two uncorrelated noise components below and above the carrier frequency always contribute to noise modulation, a factor of $\sqrt{2}$ has to be taken into account [30]. Using the RMS value of signal and noise voltage, the following equation is given
(26)$$u_{s}sin\Delta \phi_{i}\approx u_{s}\Delta \phi_{i}=\sqrt{2}u_{n}$$Using Formula (24) of the SNR, Formula (26) above could be rewritten as
(27)$$\Delta \phi_{i}=\frac{\sqrt{2}u_{n}}{u_{s}}=\sqrt{\frac{2P_{n}}{P_{s}}}=\frac{\sqrt{2}}{\sqrt{SNR}}\;.$$The minimum detectable displacement $s_{min}$ of the i-th single measuring point is then given by
(28)$$s_{min}=\frac{d}{2\pi }\Delta \phi_{i}=\frac{d}{2\pi }\frac{\sqrt{2}}{\sqrt{SNR}}\;.$$Using Formula (26) of the SNR, Formula (30) is reformulated as
(29)$$s_{min}=\frac{G_{n}d\sqrt{2\left(4k_{B}TR_{f}+e_{n}{}^{2}+{\left(i_{n}R_{f}\right)}^{2}\right)B}}{\pi^{3}R_{f}\delta_{s}K\eta \delta_{I}E_{in}A_{0}cos\frac{\theta }{2}\left(1-cos\varphi \right)}$$Considering the SNR of both measuring points, the minimum detectable strain $\varepsilon_{min}$ is multiplied by a factor of $\sqrt{2}$ as below, because the noise of both measuring points is uncorrelated.
(30)$$\varepsilon_{min}=\frac{\sqrt{2}s_{min}}{l}$$Here l is the distance between two measuring points. If $\varepsilon_{min}$ is inserted into Formula (26) instead of $s_{min}$, the minimum detectable strain $\varepsilon_{min}$ is given by
(31)$$\varepsilon_{min}=\frac{2G_{n}d\sqrt{\left(4k_{B}TR_{f}+e_{n}{}^{2}+{\left(i_{n}R_{f}\right)}^{2}\right)B}}{\pi^{3}lR_{f}\delta_{s}K\eta \delta_{I}E_{in}A_{0}cos\frac{\theta }{2}\left(1-cos\varphi \right)}\;\;.$$

### 3.4. Simulation for the Optimal Sensor Design

The specific values of all simulation parameters are shown in Table 1. With estimated values $\delta_{s}=0.5$, η=0.5, and $\delta_{I}\in \left(0.08,\;0.24\right)$, only 2~6% of the incident laser power in the image area of the photodiode on the measuring surface $A_{0}$ could be received as a carrier signal by the photodiode.

Figure 6a shows the noise-limited resolution for a bandwidth of 1 Hz in relation to the crossing angle θ between the incident laser beams. With the crossing angle θ=70.53°, the best noise-limited resolution of $3.22\times 10^{-8}\;1/\sqrt{Hz}$ could be reached. In this circumstance, the total efficiency is 2.3%.

Figure 6b shows that our new type of laser Doppler strain sensor has a better resolution limit than a conventional strain gauge. The maximum bandwidth of the laser sensor is decided by the carrier frequency. Usually, the maximum detectable signal bandwidth could be define using Carson’s bandwidth rule. Using the normal Bragg cell, which produces a carrier frequency in the high MHz-range, even the MHz strain could be measured by our sensor. The common strain gauge can usually measure strain in resonance testing machines up to approximately 10 kHz.

## 4. Experimental Setup and Results

We successfully implemented the optical structure and the necessary electronic components such as the amplifier, limiter, and power supply in a compact housing (about 300 mm×270 mm). Figure 7 presents the experimental optical structure of our strain sensor. The light beams are sketched, with the incident beams shown in red and the scatted light in yellow.

Figure 8 shows the entire experimental setup. The optical sensor and the strain gauge are marked. The directions of light beams are sketched, with incident beams shown in red and the scatted light in yellow. The optical sensor is integrated into a resonance-testing machine (POWER SWINGly 20 kN) from SincoTec GmbH. The machine produces a sinusoidal force on both ends of the aluminum sample. The optical sensor and a traditional strain gauge measure the strain from the same area, but opposite sides. The data acquisitions of both sensors are synchronized so that comparable results could be obtained.

Only a simple measurement configuration is required before assessing the strain. The optical sensor should be pointed to the specimen perpendicularly. Then, the working distance (the distance between the sensor and the specimen) should be adjusted until the carrier signal is detected. During the measurement, the signals from both photodetectors are amplified by a limiter to fit the ADC range and then sampled by a digital data acquisition card. The digital data are finally in a PC IQ-demodulation [30] and the strain is calculated. A real-time digital strain data are acquired. The temporal measurement results from both sensors are demonstrated in Figure 9a. Since the optical sensor does not measure the strain caused by preload, the red curve (results from our optical sensor) was shifted along the *Y*-axis to match the result from the strain gauge. The strain state measured by our sensor is zero at the beginning of the measurement, whereby the state measured by a strain gauge is zero at the moment when it is installed to the test object.

For this measurement, the resonance test machine produces a positive sinusoidal force as $F=Asin\left(2\pi k\right)+A_{0}$ with an amplitude A of 1000 N, a frequency k of 40 Hz, and an offset $A_{0}$ of 1650 N (in Figure 9c). Therefore, the strain signal has an offset. Roughly, both curves in Figure 9a are similar. The functionality of our optical strain sensor could be proven. However, we could also observe a little more noise with the optical sensor. Figure 9b shows the spectral information of both results. Due to the low sampling rate of 5 kHz of the strain gauge, the signal from the laser extensometer is also limited with a bandwidth of 2.5 kHz. Even though the noise floor of our optical sensor is higher than a strain gauge, the signal amplitudes at the resonance frequency are equal. These experimental results show the strain resolution of our sensor does not reach the noise-limited resolution of $3.22\times 10^{-8}\;1/\sqrt{Hz}$ from the simulation results. Figure 9d shows a maximum strain deviation of the laser Doppler extensometer compared to the conventional strain gauge under $4\times 10^{-5}$.

The laser-speckle effect makes it difficult to find two optimal measurement positions. Laser speckle, which scatters less light power into the detection aperture, reduces the signal power. The total noise remains unchanged, because the shot noise is neglected, and other noise sources are independent of the signal power. Thus, the SNR is also reduced. Compared to out-of-plane measurements, it is much more difficult to find speckles that scatter high light power into the detection aperture during the strain measurement. If there is a 30% chance to find a good speckle, which scatters much light power, with only a 9% chance a good speckle for both measuring points could be found. If the carrier signal amplitude is below the noise level of the photodetector signal, demodulation of the Doppler frequency shift or phase shift is impossible. Figure 10 shows the measurement result for such a case.

In this case, the carrier signal amplitude is below the noise level. The only noise is demodulated. However, once a “good” speckle is found, the measurement remains several hours or even days at a constantly good SNR. Small external vibrations with submillimeter displacements and temperature fluctuations present in the machine hall where we performed the strain monitoring did not affect the measurement quality. The sensor does not need to be readjusted until the specimen is reassembled. The effort to find two good speckles could be reduced by combining diversity, which is a proven technique in laser Doppler Vibrometry [31,32]. Therefore, at the next step, we will explore the technique of diversity combining for our sensor.

## 5. Conclusions and Outlook

In summary, we developed a new type of optical strain sensor with a compact structure, which we integrated into a resonance test machine from SincoTec. Our laser Doppler extensometer measures strain in real-time using the principle of differential in-plane, laser Doppler vibrometry. Unlike other optical methods such as speckle or image correlation methods, the use of the laser Doppler principle allows our sensor to have an extremely large bandwidth (up to the MHz range). Due to the efficient signal demodulation (IQ-demodulation), a real-time digital strain signal can be generated. Therefore, even the strain in a high-frequency range can be measured in real-time. A mathematic model was derived to estimate the optimal design parameters. According to the simulation results of this model, a noise-limited resolution of $3.22\times 10^{-8}\;1/\sqrt{Hz}$ can be achieved with a crossing angle θ=70.53° (the angle between two incident laser beams). The laser Doppler extensometer delivers a comparable result to a traditional strain gauge with a maximum strain error below $4\times 10^{-5}$. Our laser extensometer is capable of monitoring the strain of a probe in a resonance-testing machine but has slightly more noise. However, we see still further optimization possibilities to explore. Especially, signal diversity may improve dramatically the speckle robustness and noise level of this kind of sensor. Therefore, polarization or angle diversity, or a combination of both will be implemented in the future. We expect a significant improvement of the actual speckle problem which will enable a fast and easy adjustment of a good speckle. Our compact remote laser Doppler strain sensor has the potential to replace strain gauges in many application areas on dynamic strain measurement.

## 6. Patents

Inventor: Fangjian Wang, Christian Rembe, Robert Kowarsch. Assignee: Clausthal University of Technology. DE102018111921A1, publication of 2019.

## Figures and Tables

**Figure 1 sensors-21-00319-f001:**
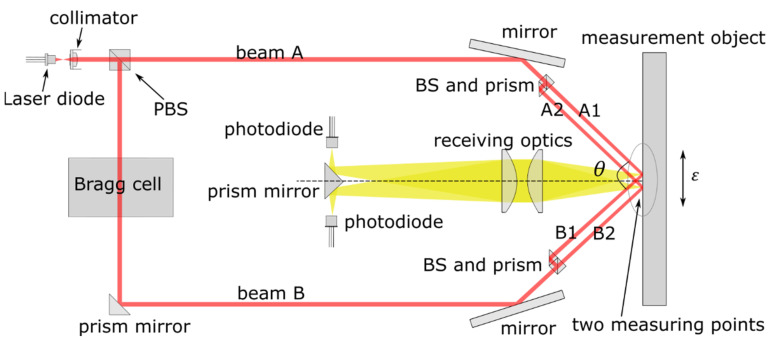
Schematic drawing of the optical setup.

**Figure 2 sensors-21-00319-f002:**
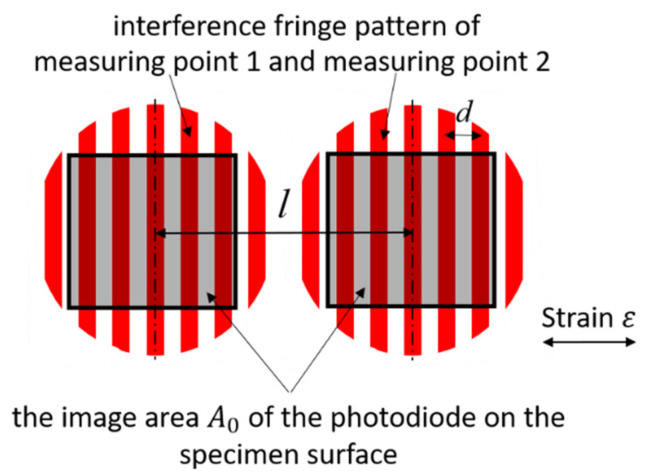
Schematic drawing of both measuring points with a fringe pattern.

**Figure 3 sensors-21-00319-f003:**
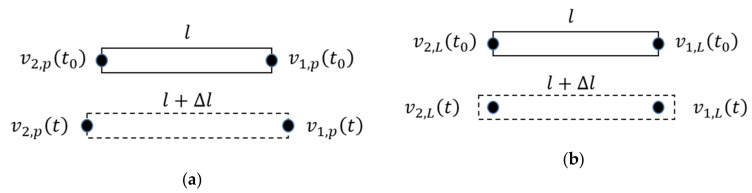
(**a**) The velocities of both endpoints of the test specimen; (**b**) The velocities measured by the laser Doppler strain sensor.

**Figure 4 sensors-21-00319-f004:**
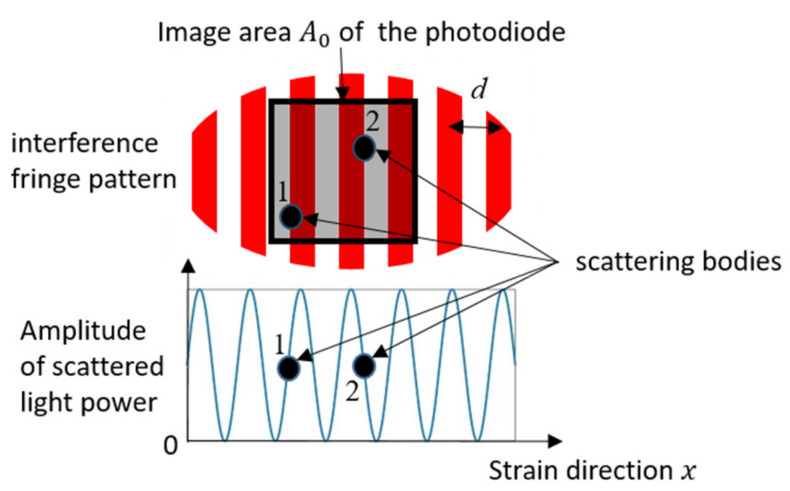
Destructive interference.

**Figure 5 sensors-21-00319-f005:**
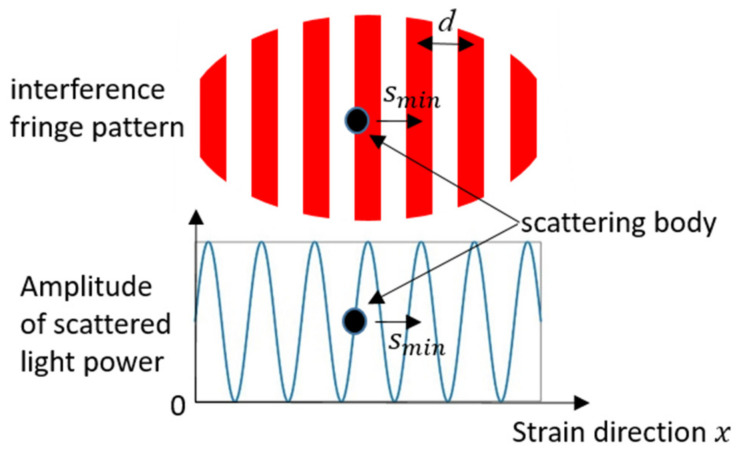
Intensity modulation, when a scattering body moves through the fringes.

**Figure 6 sensors-21-00319-f006:**
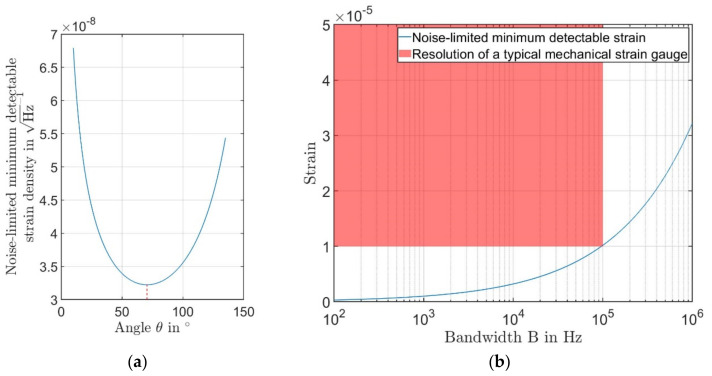
(**a**) The noise-limited minimum detectable strain in $1/\sqrt{Hz}$ in relation to the crossing angle θ; (**b**) The noise-limited minimum detectable strain in relation to the Bandwidth B.

**Figure 7 sensors-21-00319-f007:**
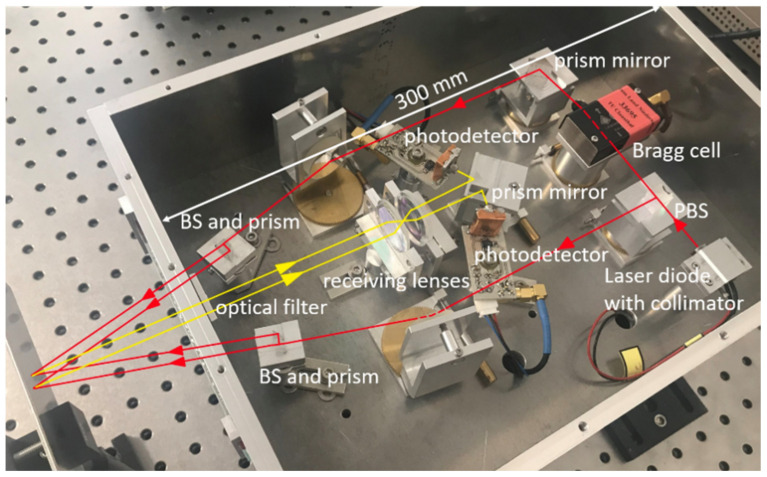
Internal optical structure of the compact strain sensor.

**Figure 8 sensors-21-00319-f008:**
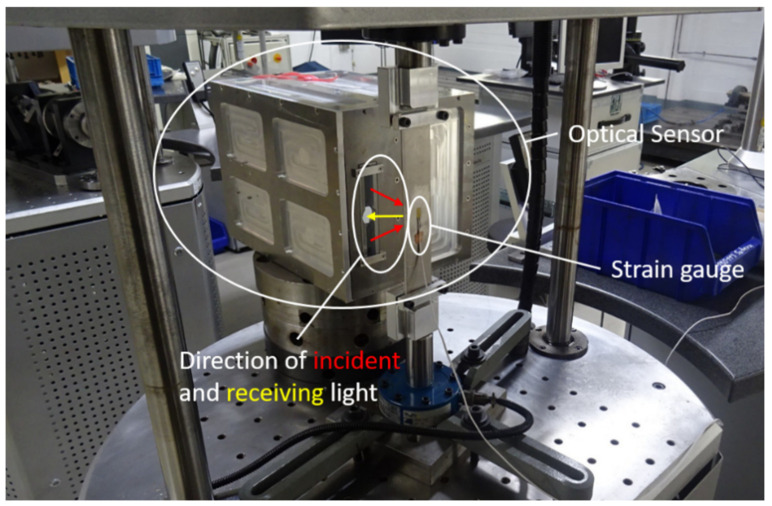
Experimental setup with the optical sensor integrated into the resonance test machine.

**Figure 9 sensors-21-00319-f009:**
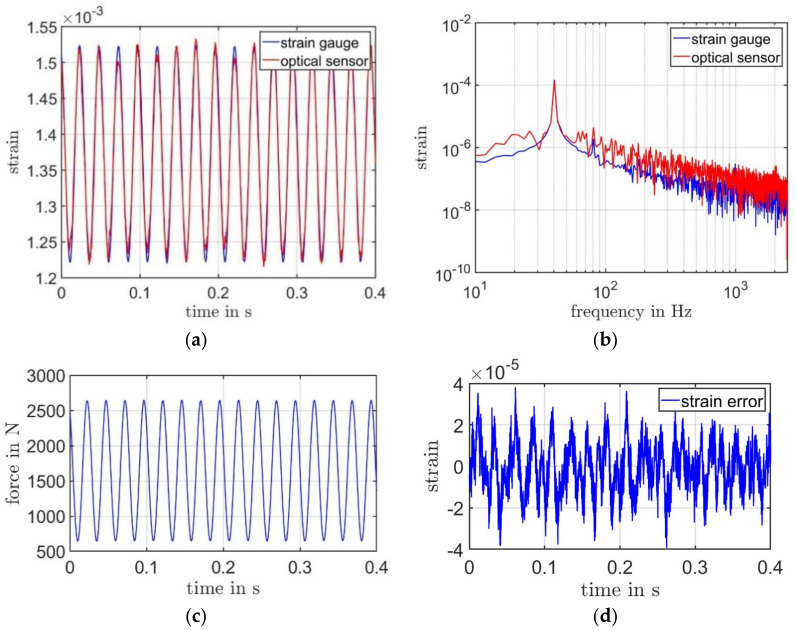
(**a**) Measurement results from the optical sensor and a traditional strain gauge. Both results are presented with a bandwidth of 2.5 kHz and a resolution bandwidth of 2.37 Hz. (**b**) The spectrum of the results from both sensors. (**c**) The sinusoidal force produced by the resonance-testing machine. (**d**) Strain deviation, the difference between results from both sensors.

**Figure 10 sensors-21-00319-f010:**
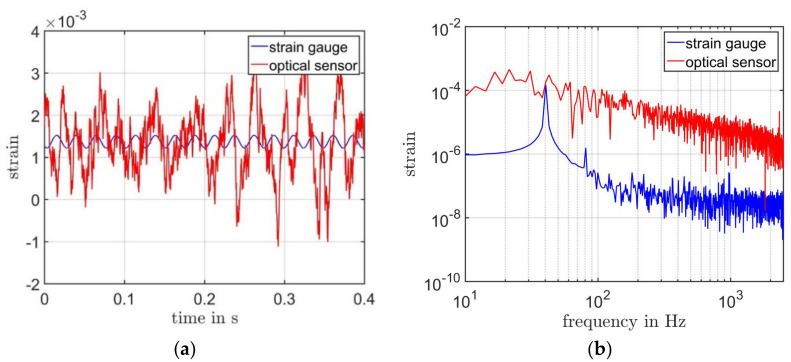
(**a**) Measurement results from the optical sensor with bad speckle and the strain gauge. Both results are presented with a bandwidth of 2.5 kHz and a resolution bandwidth of 2.37 Hz. (**b**) The spectrum of the results from both sensors.

**Table 1 sensors-21-00319-t001:** Values of all parameters.

Parameter	Value	Parameter	Value	Parameter	Value
λ	1550 nm	$E_{in}$	$3183\;W/m^{2}$	$R_{f}$	3.5 kΩ
$A_{0}$	$5.63\cdot 10^{-3}{\;mm}^{2}$	K	1.03 A/W	l	6 mm
φ	14.9°	$\delta_{s}$	0.5	η	0.5
T	293 K	$e_{n}$	$6\;nV/\sqrt{Hz}$	$i_{n}$	$1\;pA/\sqrt{Hz}$
$G_{n}$	9.6 dB				

## Data Availability

Data sharing not applicable.

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
