# Peer review of "A Contactless Laser Doppler Strain Sensor for Fatigue Testing with Resonance-Testing Machine"

_sensors, 2021, doi:10.3390/s21010319_

Round 1

Reviewer 1 Report

In this paper, a new strain sensor is developed and compared with the monitoring results of the strain gauge, the reliability of the new strain sensor is demonstrated. At the same time, it also points out the shortcomings and the direction of improvement. This article is compact in structure, detailed in content, and has considerable novelty.

The following suggestions are for the author's reference:

  1. It is suggested to add the comparison items between the new sensor and the strain gauge, such as ease of use, operation reliability, etc., to show the characteristics of the new sensor more comprehensively
  2. Can the new sensor replace some or all functions of traditional sensors such as strain gauge? What is the development direction, application scope and self-positioning of the new sensor?
  3. Some sentences are not clear or easy to cause ambiguity, so it is recommended to check them.
  4. It is recommended to indicate the position of the strain gauge and the new sensor in Figure 8.
  5. It is suggested to add the force curve in Fig. 9 to show the change of strain with force more intuitively.
  6. There is no content about the model in the conclusion. It is suggested to add a summary about the model.
  7. In this paper, the formulas for calculating the minimum measurable strain and other parameters are given. Can the author give the specific values of these parameters?
  8. The text in Figure 7 is a little fuzzy. It is recommended to enlarge the font size or adjust it appropriately.

To sum up, I suggest this article can be suggested.

Author Response

Dear Reviewer,

Thank for your suggestions. Our responses are shown in the Word file.

Best regards

Fangjian Wang

Reviewer 2 Report

Paper entitled »A contactless laser Doppler strain sensor for fatigue testing with resonance-testing machine«, presents a very interesting methodology of non-contact strain measurements and analysis.

Basically, I can only write that I really enjoyed reading the paper and I am looking forward to reading the results of your investigation in the future.

I gave some suggestions, which are more or less only cosmetic:

  • Page 2, line 84 you use NIR-laser, I would suggest since this is the first time you are mentioning that you write the full words near-Infrared Laser (NIR).
  • Page 3, the sentence: The modulation frequency is shifted by an in-plane deformation in the direction 104 perpendicular to the interference fringes of the fringe pattern on specimen’s surface, whereby the measuring point running against the interference fringes increases the Doppler-shift frequency (difference frequency between intensity modulation and 40 MHz carrier frequency) and running with the fringes reduces the Doppler-shift frequency.

The sentence is long and hard to read, I suggest the authors rewrite it.

  • I noticed some typing errors in the text, I suggest the authors another read to correct them, for example: p. 2. l. 89: Bevor; p.4. l.131: By homogeneous… Maybe use “Considering” instead “By”

I wish you all the best with your future work.

Author Response

Dear Reviewer,

Thank for your suggestions. We are pleased with your interest in our investigation. Our responses are shown in the Word file.

Best regards

Fangjian Wang

Reviewer 3 Report

This work presents a non-fiber, laser Doppler interferometer for measuring strain. The topic addressed in this research is compatible with the Journal’s aims and scope. Moreover, the manuscript is well-written and provides a comprehensive (but straightforward) analytical model of the sensor, as well as the validation by experimental analyses, confirming that the developed system yields results comparable to strain gauge extensometers. I recommend the authors to improve the paper by considering the following points:

  1. Sec. 1, lines 29 to 34: It is not usual including the page numbers of references along with the manuscript. Please check the Journal template to confirm this;
  2. Sec. 2, line 126: I suggest you clearly state that the distance l refers to the center of each measurement point, as shown in Fig. 2, and not to the position of a particular fringe;
  3. Sec. 4: Please compare the laser Doppler sensor results with the values expected in the case of assessing strain through conventional digital speckle interferometry. Is it possible to combine both methods to take advantage of the speckle noise and improve the precision of strain measurements?
  4. Please discuss the advantages and limitations of the proposed method in comparison to the available measurement systems. In particular, it would be useful if you comment on the robustness and implementation aspects, as it is apparently difficult to adjust the experimental setup for achieving a “good” speckle condition. How the system is affected by external vibration and temperature fluctuations?

Author Response

(The authors gave the same response as above.)

Round 2

Reviewer 3 Report

This work presents a non-fiber, laser Doppler interferometer for measuring strain. The authors managed to improve the manuscript following our suggestions and correctly addressed all the queries. I recommend the authors to double-check the text to correct minor grammar mistakes.

Author Response

Dear Reviewer,

Thank you for your suggestions.

We checked the total manuscript very carefully and corrected hopefully all grammar/typing errors.

Best regards

Fangjian Wang